# A Theoretical Study of Interband Absorption Spectra of Spherical Sector Quantum Dots under the Effect of a Powerful Resonant Laser

**DOI:** 10.3390/nano13061020

**Published:** 2023-03-11

**Authors:** Le Thi Dieu Hien, Le Thi Ngoc Bao, Duong Dinh Phuoc, Hye Jung Kim, C. A. Duque, Dinh Nhu Thao

**Affiliations:** 1Faculty of Physics, Hue University of Education, Hue University, 34 Le Loi Street, Hue City 530000, Vietnam; 2Center for Theoretical and Computational Physics, Hue University of Education, Hue University, 34 Le Loi Street, Hue City 530000, Vietnam; 3Faculty of Electronics, Electrical Engineering, and Material Technology, Hue University of Sciences, Hue University, 77 Nguyen Hue Street, Hue City 530000, Vietnam; 4Department of Physics, Pusan National University, Busan 46241, Republic of Korea; 5Grupo de Materia Condensada-UdeA, Instituto de Física, Facultad de Ciencias Exactas y Naturales, Universidad de Antioquia UdeA, Calle 70 No. 52-21, Medellin 050011, Colombia

**Keywords:** interband absorption spectra, dynamical coupling, spherical sector quantum dot, redshift, apical angle

## Abstract

We explore the variation of interband absorption spectra of GaAs spherical sector quantum dots (QDs) in response to a strong resonant laser, using the renormalized wave function method. Even though a spherical sector QD appears identical to a section cut from a spherical QD, it contains a controllable additional spatial parameter, the apical angle, which results in radically different wave functions and energy levels of particles, and is anticipated to exhibit novel optical properties. The obtained findings reveal that the apical angle of the dot has a considerable effect on the interband absorption spectrum. With the increase in the dot apical angle, a significant redshift of the interband absorption peaks has been identified. Increasing the pump laser detuning and dot radius yields similar results. Especially when a powerful resonant laser with tiny detuning is utilized, a dynamical coupling between electron levels arises, resulting in the formation of new interband absorption peaks. These new peaks and the former ones were similarly influenced by the aforementioned parameters. Furthermore, it is thought that the new peaks, when stimulated by a suitable laser, will produce the entangled states necessary for quantum information.

## 1. Introduction

Recently, quantum dots (QDs) have gained considerable interest [1,2,3,4,5]. The diversity in size and shape (spherical, disk-shaped, ring-shaped, ellipsoid, conical, cylindrical, and pyramidal) as well as their unique physical properties, arising from the quantum size effect, have made QDs the ideal candidates for a variety of applications. In addition to applications in traditional fields such as lasers [1], LEDs [2], medical imaging [3], and optoelectronic devices [4], QDs have applications in new areas such as quantum computing and quantum information [5,6], QD-based photonics [7], biomedical applications [8], and electrochemical biosensors [9]. Notably, they have the potential to create a novel type of flash memory, nonvolatile memory devices based on self-organized III-V QDs [10,11,12]. Furthermore, the integration of heterostructures with QDs has many benefits, and makes QDs the leading candidates for creating novel devices on mismatched substrates, for example III-V/Si, such as high-quality light-emitting HSs [13], electrically pumped lasers for Si photonics [14], and on-chip photodetectors [15]. Returning to the variety of QD shapes, it would appear that the energy spectra of QD structures can be more easily manipulated if they have more distinct geometrical properties. The cone-shaped QDs, which have been extensively studied over many years [16,17,18,19,20,21,22,23,24,25,26,27,28,29], serve as an example of such systems. The experimental research conducted by Schamp et al. [18], on the fabrication of so-called “ice cream cone” structures, based on GaAs and GaSb, is especially interesting. They appear to have developed conical QDs with a spherical cap, a structure that is mathematically beautiful and easy to analyze.

A high-intensity pump laser can couple two quantized electron states in low-dimensional semiconductor devices, causing the exciton absorption spectra to change and split [30,31]. In 1986 [30], the split, or more particularly the exciton optical Stark effect, was identified for the first time in quantum wells. Due to its potential use in ultrafast nonlinear optical devices, such as optical gating [32,33] and high-speed all-optical switching [30,34], as well as its contribution to the understanding of the interaction between photons and semiconductors [35], this separation has received considerable theoretical and experimental attention [30,31,36,37,38,39,40,41,42,43,44,45,46,47,48,49,50,51,52,53,54,55,56,57,58,59]. There are two types of separation: two-level [60] and three-level [60,61]. Several theoretical approaches, such as the renormalized wave function formulation [48], the finite difference method [49], the many-body perturbation theory [50], the photoemission theory [51], or the density matrix approach [52,53], have been utilized to study the separation thus far. Despite the fact that each of these methods has its own benefits, the first approach is usually used for the study of the three-level effect, since it provides a clear quantum-mechanical explanation of the phenomenon.

Using the renormalized wave function formulation, the interband absorption spectra of GaAs conical QDs with a spherical cap (or spherical sector QDs), under the effect of a strong resonant laser, are investigated in this paper. The experimental work of Schamp et al. [18], which studied the formation of spherical sector-like QDs based on GaAs, led to the selection of GaAs as the dot material. The paper consists of four sections: Section 2 of the paper presents the theoretical background. The numerical results and discussion of the interband absorption spectra are presented in Section 3. Lastly, we provide our conclusions in Section 4.

## 2. Theoretical Framework

### 2.1. The Wave Function and Energy Levels of Particles

Here, we investigate the spherical sector quantum dot (QD). The three-dimensional (3D) picture in Figure 1a depicts the structure as being a conical QD with a spherical cap.

We shall first determine the wave function and energy expressions of electrons and holes in a spherical sector QD with infinite confinement potential. Consider an electron that is confined in a spherical sector QD, with radius R, and the apical angle $\theta_{0}$, as shown in Figure 1b. In this instance, the confinement potential of the system is defined as follows:(1)$$V(\vec{r})=\left\{ \begin{array}{ll} 0, & r\leq R,\;\left|\theta \right|\leq \theta_{0},\\ \infty , & \mathrm{otherwhere}. \end{array}\right.$$

The assumption of the potential is purely for convenience and is not expected to change the physics picture. To be precise, the infinite barrier approximation is nearly applicable to unshelled QDs, in which the barrier is determined by the work function of electrons in a vacuum, that has a rather large value. Although spherical sector QDs are merely a piece cut from spherical QDs, the presence of a further tunable spatial parameter, the apical angle, causes a complete change to the wave functions and energy levels of the particles, which naturally changes the physical features of the system. The Schrödinger equation for the motion of the particle in the QD can be written as follows, in the spherical coordinate system (r,θ,φ):(2)$$\begin{array}{l} {}[-\frac{\hbar^{2}}{2m_{e,h}^{*}}\{\frac{1}{r^{2}}\frac{\partial }{\partial r}\left(r^{2}\frac{\partial }{\partial r}\right)+\frac{1}{r^{2}\sin\theta }\frac{\partial }{\partial \theta }\left(\sin\theta \frac{\partial }{\partial \theta }\right)\\ \;\;\;\;\;\;\;\;\;\;\;\;\;\;\;\;\;\;\;\;+\frac{1}{r^{2}\sin^{2}\theta }\frac{\partial^{2}}{\partial \varphi^{2}}\}+V\left(r,\theta ,\varphi \right)]\;\Psi \left(r,\theta ,\varphi \right)\;=E\Psi \left(r,\theta ,\varphi \right), \end{array}$$
where $m_{e,h}^{*}$ is the electron (hole) effective mass. The envelope wave functions of an electron (hole) in a spherical sector QD are obtained from Equation (2), using the method of variable separation and the wave function’s normalization condition, and they take the following form:(3)$$\Psi_{m,\nu }^{e,h}\left(\vec{r}\right)\equiv \Psi_{m,\nu }^{e,h}\left(r,\theta ,\varphi \right)=N\frac{1}{\sqrt{r}}J_{\nu +\frac{1}{2}}\left(k_{e,h}r\right)P_{\nu }^{m}\left(\cos\theta \right)e^{im\varphi },$$
where m represents the magnetic quantum number, N is the normalizing constant, $k_{e,h}=\sqrt{2m_{e,h}^{*}E/\hbar^{2}}$ and ν are analogous to the quantum numbers characterizing the radial and polar movements of electrons, respectively. Generally, it is noted that ν is not an integer number, and $\nu \neq 0\;\forall \theta_{0}$. $J_{\nu +1/2}\left(k_{e,h}r\right)$ and $P_{\nu }^{m}\left(x\right)$ are the Bessel functions of the first kind and the Legendre polynomials of the first kind, respectively. Finally, if the origin of energy is positioned at the top of the valence band, we can represent the quantization energy levels of electrons (holes) as follows:(4)$$\left\{ \begin{array}{l} E_{n}^{e}=E_{g}+\frac{\hbar^{2}\chi_{n}^{2}}{2m_{e}^{*}R^{2}},\\ E_{n}^{h}=-\frac{\hbar^{2}\chi_{n}^{2}}{2m_{h}^{*}R^{2}}, \end{array}\right.$$
where $\chi_{n}$ represents the *n*th zero (*n* = 1, 2, 3, …), corresponding to the indices ($m_{n},\nu_{n}$) of the Bessel function $J_{\nu +1/2}(k_{e,h}r)$, and obeys $\chi_{n}<\chi_{n+1}$. We can get values of ν from the boundary condition $P_{\nu }^{m}\left(\cos\theta_{0}\right)=0$, since the wave function in Equation (3) vanishes at r=R and $\theta =\theta_{0}$. The zeros are then determined by solving the equation $J_{\nu +1/2}\left(k_{e,h}R\right)=0$, using those values. The first few zeros, with the apical angle of $30^{{}^{\circ }}$, are shown in Table 1. These zeros differ significantly from the zeros of the spherical Bessel function seen in spherical QDs [62]. Additionally, these zeros are dependent on the spherical sector QDs’ apical angle as well as their dot radius.

The envelope wave functions of electrons (holes) in the spherical sector QD currently take the form:(5)$$\Psi_{n}^{e,h}\left(\vec{r}\right)\equiv \Psi_{m_{n},\nu_{n}}^{e,h}\left(\vec{r}\right)=N\frac{1}{\sqrt{r}}J_{\nu_{n}+\frac{1}{2}}\left(\frac{\chi_{n}}{R}r\right)P_{\nu_{n}}^{m_{n}}\left(\cos\theta \right)e^{im_{n}\varphi }.$$

These wave functions obviously differ from those seen in spherical QDs, which are a combination of the spherical harmonic function and the spherical Bessel function [62]. Additionally, these wave functions clearly depend on the apical angle and dot radius of the spherical sector QDs.

For the aforementioned energy levels, the time-dependent total wave functions of electrons and holes take the following form:(6)$$\left\{ \begin{array}{l} \Psi_{n}^{e}(\vec{r},t)=u_{c}\left(\vec{r}\right)\Psi_{n}^{e}\left(\vec{r}\right)e^{-\frac{i}{\hbar }E_{n}^{e}t},\\ \Psi_{n}^{h}(\vec{r},t)=u_{v}\left(\vec{r}\right)\Psi_{n}^{h}\left(\vec{r}\right)e^{-\frac{i}{\hbar }E_{n}^{h}t}, \end{array}\right.$$
where $u_{c}\left(\vec{r}\right)$ and $u_{v}\left(\vec{r}\right)$ are the periodic Bloch functions that are located close to the center of the Brillouin zone in the conduction and valence bands, respectively.

### 2.2. The Intraband Optical Transition Matrix Element

We then calculate the equation of the intraband optical transition matrix element between the two electron levels when an intensity pump laser is present. By using the Coulomb gauge, and assuming that the electromagnetic field strength is not too strong to eliminate the higher-order terms, the Hamiltonian interaction between the electron and the pump laser field can be expressed as follows:(7)$${\hat{H}}_{int}^{p}=-\frac{q}{m_{0}}\frac{A_{p}e^{-i\omega_{p}t}}{i\omega_{p}}\hat{\vec{n}}\cdot \hat{\vec{p}}=V_{p}e^{-i\omega_{p}t},$$
where we set:(8)$$V_{p}=V_{p}^{*}=-\frac{q}{m_{0}}\frac{A_{p}}{i\omega_{p}}\hat{\vec{n}}\cdot \hat{\vec{p}},$$
where, $A_{p}$ and $\omega_{p}$ represent the amplitude and frequency of the pump laser, respectively. As a result, the optical transition matrix element between the two electron levels $E_{1}^{e}$ and $E_{2}^{e}$ is defined as follows:(9)$$v_{21}=\langle u_{c}(\vec{r})\Psi_{2}^{e}(\vec{r})|{\hat{H}}_{int}^{p}|u_{c}(\vec{r})\Psi_{1}^{e}(\vec{r})\rangle .$$

By putting Equation (7) into Equation (9), we get:(10)$$v_{21}=\langle u_{c}(\vec{r})\Psi_{2}^{e}(\vec{r})|V_{p}|u_{c}(\vec{r})\Psi_{1}^{e}(\vec{r})\rangle e^{-i\omega_{p}t}\equiv V_{21}e^{-i\omega_{p}t},$$
where,
(11)$$\begin{array}{ll} V_{21} & =\langle u_{c}(\vec{r})\Psi_{2}^{e}(\vec{r})|V_{p}|u_{c}(\vec{r})\Psi_{1}^{e}(\vec{r})\rangle \\ & =-\frac{q}{m_{0}}\frac{A_{p}}{i\omega_{p}}\langle \Psi_{2}^{e}(\vec{r})|\hat{\vec{n}}\cdot \hat{\vec{p}}|\Psi_{1}^{e}(\vec{r})\rangle \\ & =\frac{q}{m_{0}}\frac{A_{p}}{i\omega_{p}}\frac{m_{e}^{*}}{i\hbar }\left(E_{2}^{e}-E_{1}^{e}\right)\langle \Psi_{2}^{e}\left(\vec{r}\right)|\hat{\vec{n}}\cdot \vec{r}|\Psi_{1}^{e}\left(\vec{r}\right)\rangle . \end{array}$$

In order to perform further calculations, we assume that the incident laser is linearly polarized along the *Ox* axis and propagated along the *Oz* axis. The intraband transition matrix element can therefore be expressed as follows:(12)$$V_{21}=\frac{q}{m_{0}}\frac{A_{p}}{i\omega_{p}}\frac{m_{e}^{*}}{i\hbar }\left(E_{2}^{e}-E_{1}^{e}\right)\int_{V}{\left(\Psi_{2}^{e}\left(\vec{r}\right)\right)}^{*}\Psi_{1}^{e}\left(\vec{r}\right)r\sin\theta \cos\varphi dV.$$

### 2.3. The Interband Absorption Spectra in the Absence of a Pump Field

In this work, we study the interband absorption spectra in spherical sector QDs (see Figure 1a). Furthermore, we will irradiate the QDs with a strong resonant laser, which is expected to couple two electron states, resulting in splitting of electron levels. For this purpose, we consider an electron confined in a spherical sector QD, of radius R, and apical angle $\theta_{0}$, surrounded by an infinite confinement potential (see Equation (1)).

To investigate the above problem, we apply a three-level system consisting of the lowest quantized energy level of the hole, and the two lowest quantized energy levels of the electron (Figure 2). Table 1 shows that $\chi_{1}$ and $\chi_{2}$ are the two lowest zeros of the Bessel function. As a result, the lowest energy level of the hole is $E_{1}^{h}$, while the two lowest energy levels of the electron are $E_{1}^{e}$ and $E_{2}^{e}$, as follows (see Equation (4)):(13)$$\left\{ \begin{array}{l} E_{1}^{h}=-\frac{\hbar^{2}\chi_{1}^{2}}{2m_{h}^{*}R^{2}},\\ E_{1}^{e}=E_{g}+\frac{\hbar^{2}\chi_{1}^{2}}{2m_{e}^{*}R^{2}},\\ E_{2}^{e}=E_{g}+\frac{\hbar^{2}\chi_{2}^{2}}{2m_{e}^{*}R^{2}}, \end{array}\right.$$
where $E_{g}$ is the band gap of the dot material, ℏ is the Planck constant, and $m_{e}^{*}$ and $m_{h}^{*}$ are the effective masses of the electron and hole, respectively. The time-dependent wave functions of the electron and hole for the above energy levels are given as (see Equation (6)):(14)$$\left\{ \begin{array}{l} \Psi_{1}^{h}(\vec{r},t)=u_{v}\left(\vec{r}\right)\Psi_{1}^{h}\left(\vec{r}\right)e^{-\frac{i}{\hbar }E_{1}^{h}t},\\ \Psi_{1}^{e}(\vec{r},t)=u_{c}\left(\vec{r}\right)\Psi_{1}^{e}\left(\vec{r}\right)e^{-\frac{i}{\hbar }E_{1}^{e}t},\\ \Psi_{2}^{e}(\vec{r},t)=u_{c}\left(\vec{r}\right)\Psi_{2}^{e}\left(\vec{r}\right)e^{-\frac{i}{\hbar }E_{2}^{e}t}, \end{array}\right.$$
where $u_{c}\left(\vec{r}\right)$ and $u_{v}\left(\vec{r}\right)$ are the periodic Bloch functions located near the center of the Brillouin zone in the conduction and valence band, respectively.

Additionally, the system is irradiated with the use of two lasers, operating simultaneously. The coupling of the electron states is provided by one, namely the pump laser. The interband absorption spectrum is identified by another, the probe laser. These laser beams are projected onto the same cross-sectional area of the QD. The laser electric fields stay in the *Oxy*-plane, since their direction of propagation is assumed to be parallel to the *Oz* axis (see Figure 1a). The form of these lasers is as follows:(15)$$\vec{E}\left(t\right)=\hat{\vec{n}}A_{j}e^{-i\omega_{j}t},$$
where $\hat{\vec{n}}$ is the unit vector along with the polarization direction of the wave, *j* = *p* (or *t*) indicates the pump laser (or the probe laser), and $A_{j}$ is the amplitude of the considered laser.

The selection rule for QD structures states that only the interband optical transition from the lowest level of hole $E_{1}^{h}$, to the lowest level of electron $E_{1}^{e}$, exists when the system is excited by a suitable probe laser that is linearly polarized (Figure 2). Therefore, the interband optical transition matrix element between these two levels is given as:(16)$$T_{11}=\langle \Psi_{1}^{e}(\vec{r},t)|{\hat{H}}_{int}^{t}|\Psi_{1}^{h}(\vec{r},t)\rangle ,$$
in which ${\hat{H}}_{int}^{t}$ describes the interaction between the electron and the probe laser field, which can be written as follows [63]:(17)$${\hat{H}}_{int}^{t}=-\frac{q}{m_{0}}\frac{A_{t}e^{-i\omega_{t}t}}{i\omega_{t}}\hat{\vec{n}}\cdot \hat{\vec{p}},$$
where q, $m_{0}$, $\hat{\vec{p}}$ are the charge, the bare mass, and the momentum operator of the electron, respectively; and $A_{t}$ and $\omega_{t}$ are the amplitude and the frequency of the probe laser, respectively. By substituting Equations (14) and (17) into Equation (16), we may finally derive the interband optical transition matrix element as:(18)$$T_{11}=-\frac{q}{m_{0}}\frac{A_{t}p_{cv}}{i\omega_{t}}e^{\frac{i}{\hbar }\left(E_{1}^{e}-E_{1}^{h}-\hbar \omega_{t}\right)t},$$
where $p_{cv}$ is the polarization matrix element between the conduction and the valence band, as determined by:(19)$$p_{cv}=\langle u_{c}\left(\vec{r}\right)\left|\hat{\vec{n}}\cdot \;\hat{\vec{p}}\right|u_{v}\left(\vec{r}\right)\rangle .$$

In accordance with Fermi’s golden rule, the expression for the transition rate between the lowest level of the hole and the lowest level of the electron, in the absence of the pump laser, is provided by [64]
(20)$$W_{0}=\frac{2\pi }{\hbar }{\left|T_{11}\right|}^{2}\delta \left(E_{1}^{e}-E_{1}^{h}-\hbar \omega_{t}\right).$$

The delta function, δ, in Equation (20) can be replaced by a narrow Lorentzian by means of [65]:(21)$$\delta \left(E_{1}^{e}-E_{1}^{h}-\hbar \omega_{t}\right)=\frac{\Gamma }{\pi \left\{{\left(E_{1}^{e}-E_{1}^{h}-\hbar \omega_{t}\right)}^{2}+\Gamma^{2}\right\}},$$
where Γ is the phenomenological linewidth of the absorption peak.

By substituting Equations (18) and (21) into Equation (20), we get the final expression of the transition rate, in the following form:(22)$$W_{0}=\frac{2}{\hbar }{\left(\frac{qA_{t}p_{cv}}{m_{0}\omega_{t}}\right)}^{2}\frac{\Gamma }{{\left(E_{1}^{e}-E_{1}^{h}-\hbar \omega_{t}\right)}^{2}+\Gamma^{2}}.$$

### 2.4. The Interband Absorption Spectra in the Presence of a Pump Field

When the system is driven by a high-intensity pump laser, which resonates with the energy spacing between the first two quantized energy levels of the electron, the electron is in a superposition state, represented by a renormalized wave function of the following form:(23)$$\begin{array}{l} \Psi_{mix}(\vec{r},t)=\frac{1}{2\Omega_{R}}\left(\alpha_{1}e^{-\frac{i}{\hbar }E_{1}^{e-}t}+\alpha_{2}e^{-\frac{i}{\hbar }E_{1}^{e+}t}\right)u_{c}\left(\vec{r}\right)\Psi_{1}^{e}(\vec{r})\\ \;\;\;\;\;\;\;\;\;\;\;\;\;\;\;-\frac{V_{21}}{2\hbar \Omega_{R}}\left(e^{-\frac{i}{\hbar }E_{2}^{e-}t}-e^{-\frac{i}{\hbar }E_{2}^{e+}t}\right)u_{c}\left(\vec{r}\right)\Psi_{2}^{e}(\vec{r}), \end{array}$$
where,
(24)$$\left\{ \begin{array}{l} \Omega_{R}={\left[\frac{\Delta \omega^{2}}{4}+\frac{|V_{21}{|}^{2}}{\hbar^{2}}\right]}^{1/2},\\ \alpha_{1}=\Omega_{R}-\frac{\Delta \omega }{2},\\ \alpha_{2}=\Omega_{R}+\frac{\Delta \omega }{2},\\ \Delta \omega =\omega_{p}-\frac{E_{2}^{e}-E_{1}^{e}}{\hbar }, \end{array}\right.$$
in which $V_{21}$ is the intraband optical transition matrix element between the first two lowest quantized levels of the electron (see Equation (12)); and ℏ△ω is the detuning between the first two electron levels and the photon energy of the pump laser, $\hbar \omega_{p}$, that must satisfy the following condition:(25)$$\hbar △\omega \ll \hbar \omega_{p}\ll E_{g}.$$

From Equation (23), we see that the quantized energy spectrum of the electron in the superposition state consists of four levels, $E_{1}^{e\pm }$ and $E_{2}^{e\pm }$, that are separated from $E_{1}^{e}$ and $E_{2}^{e}$ levels, respectively (Figure 3). These separation energy levels are given by:(26)$$\left\{ \begin{array}{l} E_{1}^{e-}=E_{1}^{e}-\hbar \alpha_{2},\\ E_{1}^{e+}=E_{1}^{e}+\hbar \alpha_{1}, \end{array}\right.$$
and
(27)$$\left\{ \begin{array}{l} E_{2}^{e-}=E_{2}^{e}-\hbar \alpha_{1},\\ E_{2}^{e+}=E_{2}^{e}+\hbar \alpha_{2}. \end{array}\right.$$

From Equations (24), (26) and (27), we have:(28)$$\left\{ \begin{array}{l} E_{1}^{e+}-E_{1}^{e-}=2\hbar \Omega_{R},\\ E_{2}^{e+}-E_{2}^{e-}=2\hbar \Omega_{R}. \end{array}\right.$$

Now we compute the matrix element for the optical transition between the hole state and the electronic superposition state, under the effect of a strong pump laser, which is described by:(29)$$T_{mix,0}=\langle \Psi_{mix}(\vec{r},t)|{\hat{H}}_{int}^{t}|\Psi_{1}^{h}(\vec{r},t)\rangle .$$

By substituting Equations (14), (17) and (23) into Equation (29), and applying the selection rule for the optical transition in the QD structure, the expression in Equation (29) can be rewritten as:(30)$$T_{mix,0}=-\frac{qA_{t}p_{cv}}{m_{0}i\omega_{t}}\left(\frac{\alpha_{1}}{2\Omega_{R}}e^{\frac{i}{\hbar }\left(E_{1}^{e-}-E_{1}^{h}-\hbar \omega_{t}\right)t}+\frac{\alpha_{2}}{2\Omega_{R}}e^{\frac{i}{\hbar }\left(E_{1}^{e+}-E_{1}^{h}-\hbar \omega_{t}\right)t}\right).$$

Therefore, the probability of the interband optical transition between the hole state and the electronic superposition state in the presence of an intensive pump laser is determined as:(31)$$W=\frac{2\pi }{\hbar }{\left(\frac{qA_{t}p_{cv}}{m_{0}\omega_{t}}\right)}^{2}\left[{\left(\frac{\alpha_{1}}{2\Omega_{R}}\right)}^{2}\delta \left(E_{1}^{e-}-E_{1}^{h}-\hbar \omega_{t}\right)+{\left(\frac{\alpha_{2}}{2\Omega_{R}}\right)}^{2}\delta \left(E_{1}^{e+}-E_{1}^{h}-\hbar \omega_{t}\right)\right].$$

By substituting the δ functions in Equation (31) with a narrow Lorentzian function, similar to those in Equation (21), the equation may be made simpler. From this, we can derive the final expression of the transition rate under the effect of a strong pump laser as:(32)$$W=\frac{2}{\hbar }{\left(\frac{qA_{t}p_{cv}}{m_{0}\omega_{t}}\right)}^{2}[{\left(\frac{\alpha_{1}}{2\Omega_{R}}\right)}^{2}\frac{\Gamma }{{\left(E_{1}^{e-}-E_{1}^{h}-\hbar \omega_{t}\right)}^{2}+\Gamma^{2}}+{\left(\frac{\alpha_{2}}{2\Omega_{R}}\right)}^{2}\frac{\Gamma }{{\left(E_{1}^{e+}-E_{1}^{h}-\hbar \omega_{t}\right)}^{2}+\Gamma^{2}}].\;$$

We were unable to identify the explicit relationship of the transition rate on the dot radius and apical angle by looking at Equation (32). The transition rate, however, depends on several different factors such as $\Omega_{R}$, $\alpha_{1}$, $\alpha_{2}$, and Δω, all of which have a connection to the energy levels and wave functions of the particles. As mentioned in the previous sections, the energy levels and wave functions of particles are functions of the dot radius and apical angle. As a result, such variables also affect the transition rate.

## 3. Numerical Results and Discussions

In this section, we will study the optical transition rates in order to examine the interband absorption spectra in GaAs conical QDs with a spherical cap under the action of a strong pump laser. Our calculations are based on the following parameters: $m_{e}^{*}=0.067m_{0}$ and $m_{h}^{*}=0.51m_{0}$, which are the effective masses of electron and hole [66], respectively (where $m_{0}$ is the free electronic mass); $E_{g}$ = 1.424 eV is the band gap of GaAs semiconductor [67]; the amplitude of the pump laser is $A_{p}=6\times 10^{6}\;V/m$; and the linewidth is Γ=0.1 meV.

It is worth noting that the main purpose of this research is to address the excitonic optical Stark effect in spherical sector QDs. For the time being, however, we prefer working with the interband absorption spectra, over the exciton absorption spectra, for the following reasons. In principle, under the proper conditions, we can investigate the absorption spectra of QDs caused by the interband transition. This process can produce either an exciton or an electron–hole pair, depending on whether there is Coulomb interaction between the electron and the hole. The energy of an electron–hole pair in a QD is calculated by adding the quantized energy levels of the electron and hole, as well as the bandgap of the dot material. In a strong confinement regime, like in this work, the exciton wave functions can be treated as the product of the electron and hole wave functions. Meanwhile, the exciton energy level is equal to the difference of the energy of the electron–hole pair and the exciton binding energy, which results from the electron–hole Coulomb interaction. Typically, the exciton binding energy in GaAs quantum structures is several tens of meV [68,69,70,71], which is much lower than the quantized energies of the electron and hole. As a result, the issue with excitons can be handled similarly to the issue with an electron–hole pair, and the exciton absorption spectra are comparable to the interband absorption spectra.

First, we plot the dependence of the interband absorption spectra, in the absence of the pump laser, on the size of the QD, Figure 4. In this situation, we detect a single peak in the absorption spectra as well as a redshift. The absorption photon has a longer wavelength, or a lower frequency, as the dot radius grows. This dependence is well-known, as the consequence of the size effect of quantum structures.

Next, we calculate the dependence of the interband absorption spectra, in the absence of the pump laser, on the apical angle of the QD, Figure 5. Here, we also observe a single absorption peak in the spectra. We see that when the apical angle expands, the absorption peak shifts to the lower energy region, giving us the same redshift as in the earlier study. This result makes sense, because the energy levels of the carriers decrease as the apical angle rises, bringing them closer together. Therefore, the QD only has to absorb a low-energy photon to stimulate the optical transition from the hole level to the lowest level of the electron. This shift happens more quickly if the apical angle is smaller, in particular. It is really an interesting point. It appears that the dot apical angle, together with its dot size, has a significant impact on the interband absorption spectra. One of the novel findings of this research, and what sets it apart from earlier studies [62,72], is the substantial apical angle dependency of the interband absorption spectra.

From this point forward, we focus on the scenario in which the QD is irradiated by a powerful pump laser. Figure 6 shows the interband absorption spectra with dot radius R=50 Å and the apical angle $\theta_{0}=60^{\circ }$, in both the scenario where the pump laser is operating and in the absence of it. As shown in Figure 4 and Figure 5, there is just one peak (dashed line) in the spectra when there is no pump laser. However, when there is an intense pump wave, that is resonant with two levels of the electron, the spectra show two new peaks (solid line), that are symmetrically placed on either side of the initial peak (see the left side of the diagram in Figure 3).

The following is the explanation for the formation of the new peaks. When there is no pump laser present, at the beginning of the investigation, two initial electron levels $E_{1}^{e}$ and $E_{2}^{e}$ provide for four possible electron states. These two initial levels combine to form a single level in the presence of a high-power resonant pump laser, eliminating the possibility of two allowed electronic states. Therefore, each level, $E_{1}^{e}$ and $E_{2}^{e}$, must be divided into two sublevels, $E_{1}^{e\pm }$ and $E_{2}^{e\pm }$, in order to preserve the number of allowed electronic states (see the right side of the schematic in Figure 3). However, due to the selection rule in the QD structure, we only see two peaks in the absorption spectra, that correspond to two interband transitions, from the hole level $E_{1}^{h}$ to two electron sublevels $E_{1}^{e-}$ and $E_{1}^{e+}$, not from the hole level to two sublevels, $E_{2}^{e-}$ and $E_{2}^{e+}$ (Figure 3). For ease of understanding, the new peak, that has an energy greater than the initial peak, is referred to as the “higher energy peak”, and the other is referred to as the “lower energy peak”.

The formation of the two new peaks under the influence of a strong pump laser, may have practical significance. From Equation (28) it can be seen that the energy difference between the two splitting levels, $E_{1}^{e+}$ and $E_{1}^{e-}$, is double the Rabi energy, $\hbar \Omega_{R}$. Therefore, if we use a probe pulse laser with a linewidth greater than twice the Rabi energy, we can simultaneously excite two interband transitions from the hole level to these two splitting levels. Additionally, since only a laser is used to induce these two transitions, and the Rabi energy is extremely low, the two electron–hole pair states produced by these simultaneous transitions can be thought of as coherent and entangled states. This fact actually might be useful for quantum information.

We created Figure 7 to examine the effect of the pump laser’s detuning on the interband absorption spectra in the previous QD. By comparing with Figure 6, we can see that, as the detuning rises, the higher energy peak tends to move closer to the initial peak’s location (denoted by the white line), with a greater height. In contrast, the lower energy peak has a lower height, and is farther away from the initial peak. Particularly, the lower energy peak nearly vanishes when the detuning is too large, as in the case of ℏΔω=1.0 meV or larger, which means that the splitting of the electron level does not take place. Additionally, when the detuning grows, these new peaks have a tendency to experience a redshift toward the long wavelength region.

Figure 8 illustrates the interband absorption spectra as a function of dot size, under the effect of the pump laser. As the dot radius increases, it can be observed that the resonant peak positions shift towards the lower energy region, as shown in Figure 4. The physical explanation for this feature is that, when R increases, the energy levels of electrons and holes decrease, bringing them closer together. On the other hand, when the radius is increased and the apical angle is fixed, the QD’s volume will rise ($V=\frac{2\pi }{3}R^{3}(1-\cos\theta_{0})$), which causes the binding energy between the carriers to decrease—that is, the weaker the confinement, the lower the binding energy. Therefore, in order to excite the optical transition from the hole level to the splitting levels of the electron, the system just needs to absorb a photon with lower energy.

Next, we examine how the apical angle affects the interband absorption spectra when a pump laser is present. Figure 9 shows that, as the angle is increased, the position of the peaks tends to move toward the lower energy regions, or the regions with longer wavelengths, as we saw in Figure 5. Additionally, at large angles, the shift happens more gradually. This characteristic results from the fact that the volume of the spherical sector QD increases as the apical angle rises, but the dot radius stays fixed, weakening the confinement strength. Therefore, the transitions from the hole level to the splitting levels of the electron can be excited using just low-energy photons. Combining Figure 8 and Figure 9, we see that increasing the dot radius or widening the angle causes the interband absorption spectra to redshift.

The effect of the pump laser’s amplitude, $A_{p}$, on the interband absorption spectra, is seen in Figure 10. As can be seen from the graph, the height of the two new peaks does not change significantly for any value of the amplitude, but their separation increases as it grows. The following is an explanation. According to Equation (12), increasing the amplitude causes the value of the matrix element $V_{21}$ to rise, which raises the Rabi frequency, $\Omega_{R}$ (see Equation (24)). Additionally, the energy interval between the two splitting levels, $E_{1}^{e+}$ and $E_{1}^{e-}$, is larger as the Rabi frequency grows, because this energy difference is twice the Rabi energy, $\hbar \Omega_{R}$ (see Equation (28)). Consequently, the two splitting peaks move farther apart as the amplitude grows.

We noticed a similarity in graph form when comparing the interband absorption spectra of spherical sector QDs to those of spherical QDs [62] and prolate ellipsoidal QDs [72]. However, the differing forms of various QDs cause changes in the energy spectra and wave functions of electrons (holes). Hence, as seen in Figure 11, these quantitative differences result in clearly distinct interband absorption spectra in the respective structures. While having the same radius and detuning as the spherical QD, the resonant peak position of the spherical sector QD is placed in a higher energy range, as depicted in this figure. Since its volume is always smaller than that of the spherical QD, this results in a strengthening of the binding energy among the carriers in the spherical sector QD. Thus, the interband transition requires a photon with a higher energy. In addition to modifying the dot radius, as in a spherical QD or a prolate ellipsoidal QD, it is also possible to change the optical Stark effect, by adjusting the apical angle of a spherical sector QD. Hence, the spherical sector QD may be revealed to be a more suitable subject for studying the optical Stark effect.

## 4. Conclusions

In this study, we theoretically investigated the interband absorption spectra of GaAs spherical sector QDs under the effect of a powerful resonant pump laser. The results demonstrate that the apical angle, another tunable spatial feature of these QDs, has a significant effect on the particles’ wave functions and energy levels, leading to a significant shift in their interband absorption spectra. When the apical angle of the dot increases, the interband absorption peaks display a significant redshift. Increasing the pump laser detuning and dot radius provides comparable results. In addition, depending on the laser detuning, the operation of the powerful resonant laser may result in the splitting of the initial interband absorption peak into two new ones. Furthermore, if these new peaks are generated by a suitable laser, they may yield entangled states, which opens the door for the application of this phenomenon in quantum information and related fields. Because spherical sector QDs have one more adjustable spatial parameter than other QD structures, it may be easier to modify their optical properties. We expect that the achieved results will contribute to the progress of optoelectronic devices.

## Figures and Tables

**Figure 1 nanomaterials-13-01020-f001:**
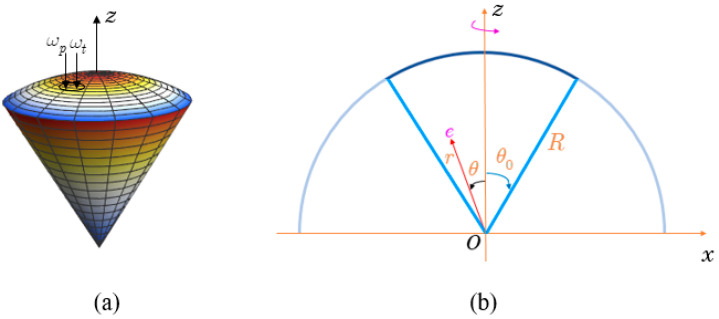
(**a**) Sketch of a three-dimensional conical QD with a spherical cap; (**b**) schematic image of the conical QD with a spherical cap, in the *y* = 0 projection [28].

**Figure 2 nanomaterials-13-01020-f002:**
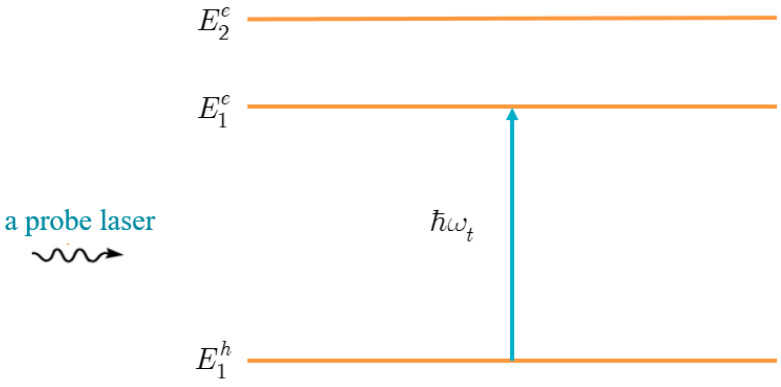
Schematic view of a three-level system, without the presence of a pump laser.

**Figure 3 nanomaterials-13-01020-f003:**
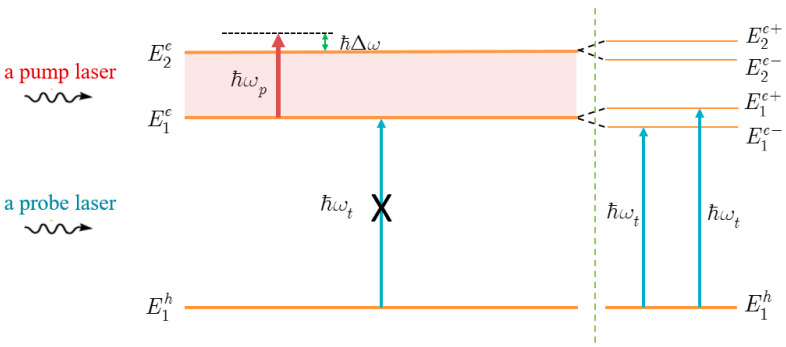
The splitting diagram of electron energy levels caused by a strong pump laser, resonant with two electron levels.

**Figure 4 nanomaterials-13-01020-f004:**
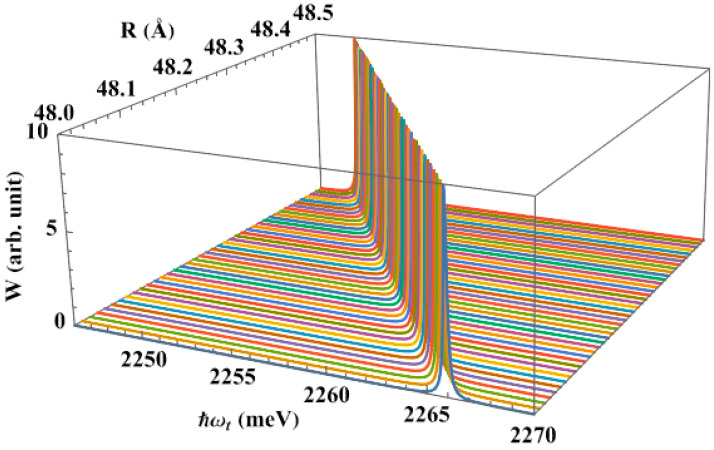
The interband absorption spectra as a function of dot radius, with the apical angle $\theta_{0}=60^{\circ }$, in the absence of the pump laser.

**Figure 5 nanomaterials-13-01020-f005:**
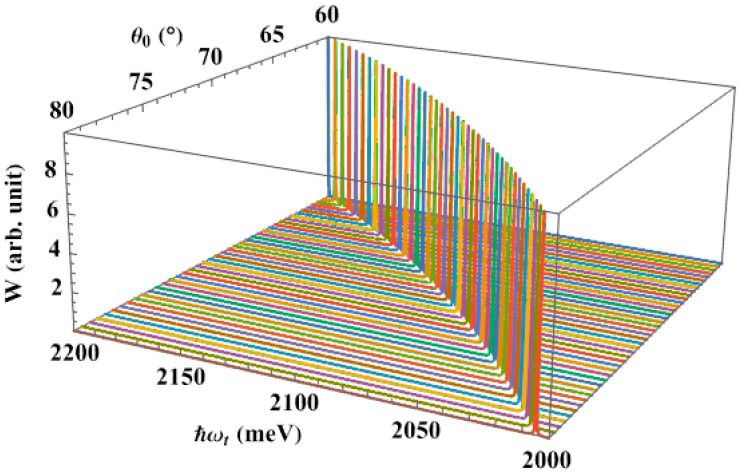
The interband absorption spectra as a function of the apical angle, with dot radius R=50 Å, in the absence of a pump laser.

**Figure 6 nanomaterials-13-01020-f006:**
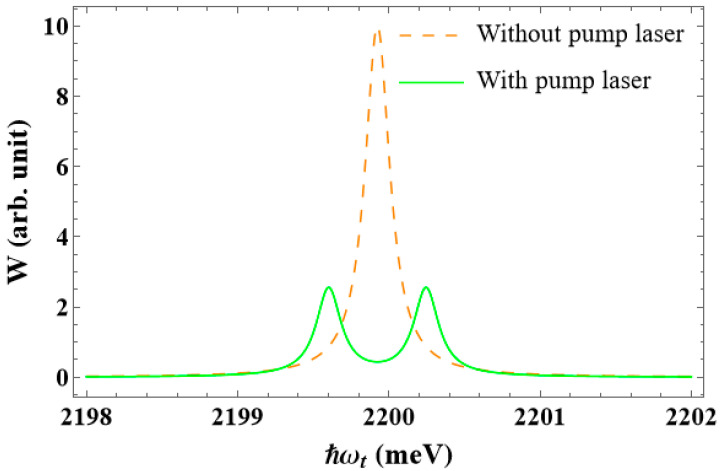
The interband absorption spectra without a pump laser, and with a pump laser that has the detuning ℏΔω=0 meV.

**Figure 7 nanomaterials-13-01020-f007:**
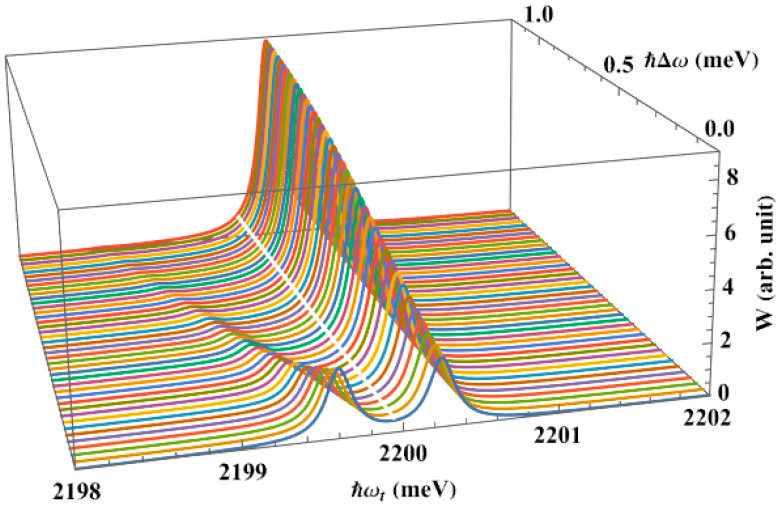
The interband absorption spectra in the presence of the pump laser, as a function of the pump laser’s detuning, with R=50 Å and $\theta_{0}=60^{\circ }$.

**Figure 8 nanomaterials-13-01020-f008:**
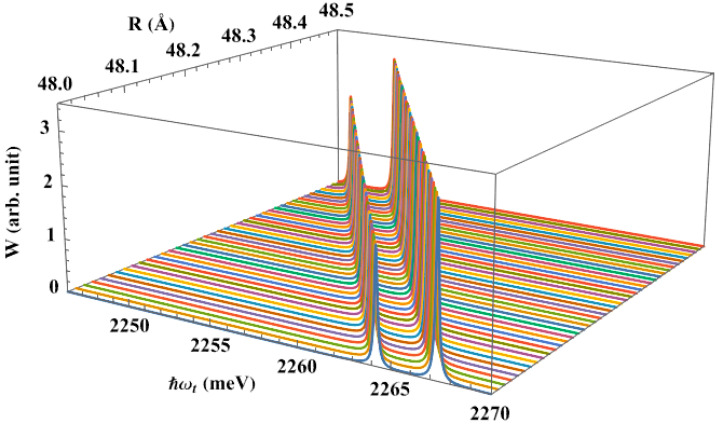
The interband absorption spectra, in the presence of the pump laser, as a function of the dot size, with $\theta_{0}=60^{\circ }$, ℏΔω=0.3 meV, and $A_{p}=30\times 10^{6}\;V/m$.

**Figure 9 nanomaterials-13-01020-f009:**
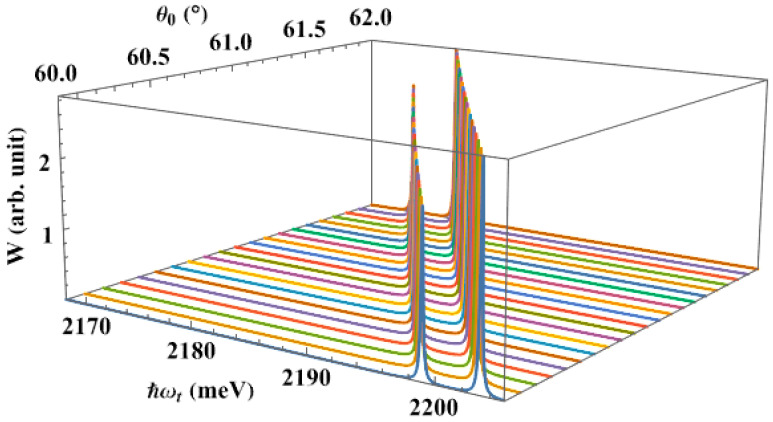
The interband absorption spectra, in the presence of the pump laser, as a function of the apical angle, with R=50 Å, ℏΔω=0.3 meV, and $A_{p}=40\times 10^{6}\;V/m$.

**Figure 10 nanomaterials-13-01020-f010:**
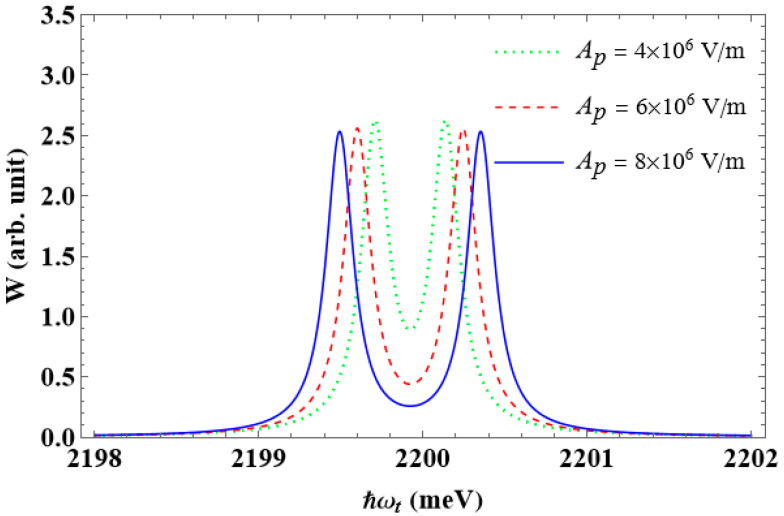
The dependence of the interband absorption spectra on the pump laser’s amplitude, $A_{p}$, with R=50 Å, $\theta_{0}=60^{\circ }$, and ℏΔω=0 meV.

**Figure 11 nanomaterials-13-01020-f011:**
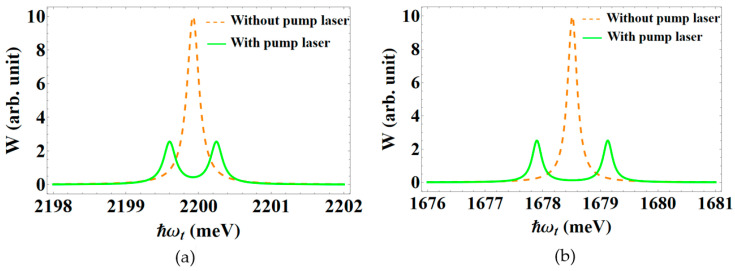
The interband absorption spectra in: (**a**) a conical QD with a spherical cap, with $\theta_{0}=60^{\circ }$; and (**b**) a spherical QD [62], with the same radius R=50 Å, and the pump laser’s detuning, ℏΔω = 0 meV.

**Table 1 nanomaterials-13-01020-t001:** The first few zeros of the Bessel function, when the apical angle is $\theta_{0}=30^{{}^{\circ }}$.

*n*	$\left(m_{n},\nu_{n}\right)$	$\chi_{n}$
1	(0, 4.08369)	8.28148
2	(1, 6.8354)	11.4695
3	(2, 9.37328)	14.3338
4	(0, 10.0386)	15.0764
5	(1, 12.9083)	18.2505
6	(2, 15.6154)	21.2108
7	(0, 16.0248)	21.6562
8	(1, 18.9364)	24.8094
9	(2, 21.721)	27.8050

## Data Availability

No new data were created or analyzed in this study. Data sharing is not applicable to this article.

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
