# Peer review of "A Theoretical Study of Interband Absorption Spectra of Spherical Sector Quantum Dots under the Effect of a Powerful Resonant Laser"

_nanomaterials, 2023, doi:10.3390/nano13061020_

Round 1

Reviewer 1 Report

The manuscript entitled "The excitonic absorption spectra under the action of a strong resonant laser in spherical sector quantum dots" by Le Thi Dieu Hien, Le Thi Ngoc Bao, Duong Dinh Phuoc, Hye Jung Kim, Carlos A. Duque, Dinh Nhu Thao is an original research paper devoted to a theoretical study of the effect of laser irradiation on the excitonic absorption spectra of quantum dots.

First, I would like to draw attention to the paper of the authors published in the Journal of Nanomaterials (Hindawi) in 2021 (Volume 2021, Article ID 5586622; https://doi.org/10.1155/2021/5586622) which looks very close to the manuscript under review (formulae, figures, conclusions etc.). The novelty of the study should be noticed more clearly.

Second, the authors claimed they study the special type of spherical sector quantum dots, but they did not give any expression describing the dependence of the transition rate W on apical angle {theta}_0, and W on QD radius R. So, the obtained results and conclusions seem to be unjustified at all.

Also, I would recommend correcting English by a native speaker.

Reviewer 2 Report

In their article the authors study the properties of a three level quantum dot in the presence of two laser fields coupling the different levels.

I find the article interesting and timely, and the methods clearly described. The physics is sound. I think the article should be published almost in its present form. 

I think that the following changes could improve the quality of the presentation:

1) I think that the first picture of the appendix should be moved to the main text. In general I think that the appendix is crucial for the understanding of the article and it might be worth to incorporate it completely in the main text, but this is a matter of taste.

2) I am not specifically working in the field of the article, so I am not sure, but as far as I understand the novel results are the ones about the quantum dot, and not the ones about the general framework presented in Sec.2. If this is the case, this fact should be stated clearly.

3) I think that a honest list of assumptions/some details about the regime in which the model is valid should be included at the level of the introduction.

Reviewer 3 Report

The work is devoted to the theoretical consideration of exciton effects in the spherical sector QD under the action of laser radiation. There are a number of significant remarks to the paper.

  1. It is not clear from the abstract and title that the study is theoretical (numerical simulation, in my opinion, is also a theoretical consideration). From the first lines of the abstract, one gets the impression that the work is experimental, that is not true.
  2. The relevance of the study of these objects is not sufficiently substantiated. To justify the relevance of the study, the authors cite only 2 references [6,7] without any additional explanations. An analysis of these works showed that in them QDs in the form of a spherical sector are considered exclusively theoretically. The authors of [6] refer to the experimental work [JLMN-Journal of Laser Micro/Nanoengineering 1, 3 (2006)], where the preparation of nanohills on amorphous Ge and Si surfaces is considered. Alas, the form of these objects not similar with the form considered by the authors of this work. The same can be said about the objects considered in the experimental work [Phys. stat. Sol. 4, 3066 (2007)], which is referred to by the authors of [6], where spherical QDs formed at the top of a conical whisker are considered. The authors of [7] refer to the experimental work [J. Appl. Phys. 77 (1995) 447–462], which discusses the growth and properties of conical nanowhiskers. Unfortunately, these objects have little in common with QDs in the form of a spherical sector, since the longitudinal dimensions of such nanowhiskers are about 1 μm, which excludes the occurrence of quantum confinement effects when charge carriers move along the structure axis. In addition, the authors of [7] refer to the experimental work [Appl. Surf. sci. 253 (2007) 6326–6329], which considers the formation of new objects, the so-called “ice cream cone” structures based on GaAs and GaSb. Fortunately, these objects are as close as possible to those considered in this paper. Alas, the authors did not reflect this work in the introduction. Also, there is practically no information on the preparation of such structures based on III-N compounds.
  3. As it clear from the text, the authors consider electron and hole states separately. At the same time, the authors claim that their study is devoted to the consideration of phenomena associated with an exciton localized in a QD. In this case, such a separate consideration of the states of charge carriers is not correct, since it does not take into account the Coulomb interaction between an electron and a hole.

Thus, the comments made cast doubt on the relevance of the study and the adequacy of the proposed methods. I believe that this paper cannot be published in its present form in the open press.

Round 2

Reviewer 1 Report

No doubt the authors have done their best to improve the paper. The revised version of the manuscript is a completed and well-organized research paper. I only recommend authors to add some recent references on a review on quantum dots properties and applications:

1. Mohammad Ali Farzin, Hassan Abdoos. A critical review on quantum dots: From synthesis toward applications in electrochemical biosensors for determination of disease-related biomolecules, Talanta, 224, 121828 (2021); https://doi.org/10.1016/j.talanta.2020.121828.

2. A.I. Arzhanov, A.O. Savostianov, K.A. Magaryan, K.R. Karimullin, A.V. Naumov. Photonics of semiconductor quantum dots: applied aspects, Photonics Russia, 16(2) 96 (2022); doi: 10.22184/1993-7296.FRos.2022.16.2.96.112.

3. Nayab Azam, Murtaza Najabat Ali, Tooba Javaid Khan. Carbon Quantum Dots for Biomedical Applications: Review and Analysis, Frontiers in Materials, 8, 700403 (2021); https://doi.org/10.3389/fmats.2021.700403.

4. Yasuhiko Arakawa, Mark J. Holmes. Progress in quantum-dot single photon sources for quantum information technologies: A broad spectrum overview, Applied Physics Reviews, 7, 021309 (2020); https://doi.org/10.1063/5.0010193.

The article can be accepted for publication in Nanomaterials after the minor revision.

Reviewer 3 Report

The authors responded to the main comments on the papere and dispelled my doubts about the value of the work. However, there are a number of minor comments, the consideration of which will make the work more interesting for a wide range of readers.

  1. The authors refer to experimental works discussing the production of «ice cream cone» shape QDs from GaAs and GaSb. In this regard, the authors choose GaAs/AlGaAs QDs of the corresponding shape as the object under study in present paper. At this stage, it is not entirely clear whether the authors plan to study buried QDs or core-shell QDs. However, the authors then state that the gap between the bands at the heterojunction is 585 meV, which is so large that one can accept the barriers as infinite. Here I will allow myself to disagree with the authors, since such a height of the energy barrier is not large enough to consider it infinite without loss of accuracy in the calculation of the energy spectrum. For example, if we compare self-assembled buried QDs formed in InAs/GaAs and InAs/AlAs heterosystems, then, with approximately equal sizes and compositions, InAs/AlAs QDs are characterized by a noticeably higher optical transition energy (1.4–1.9 eV) than InAs/AlAs/ GaAs QD (1-1.4 eV). This means that a change in the barrier height from GaAs to AlAs has a significant effect on the QD energy spectrum. Therefore, I would recommend the authors not to mention the AlGaAs barrier, but write about GaAs (or GaSb) QDs. The infinite barrier approximation is much better suited to unshelled QDs, since in this case the barrier determined by the work function of electrons in vacuum is taken into account.
  2. It is strange that in equation (2) there is no explicit potential V(r).
  3. For a complete review of the areas of application of QDs, I would recommend the authors to highlight in the introduction such a novel industry as the creation of non-volatile memory elements based on self-organized III-V QDs. I advise you to read the works: 1) Semicond. sci. Technol. 2011, 26, 014026. https://doi.org/10.1088/0268-1242/26/1/014026; 2) Phys. Status Solidi B 2016, 253, 1869. https://doi.org/10.1002/pssb.201600274; 3) Nanomaterials 2022, 12, 3794. https://doi.org/10.3390/nano12213794
  4. It is also worth paying readers' attention in the introduction to one of the main advantages of heterostructures with QDs: complete 3D localization of charge carriers in QDs effectively protects such structures from the negative effect of growing defects and makes QDs the main candidates for creating light-emitting devices and photodetectors on mismatched substrates, for example III-V/Si. I advise you to get acquainted with the works: 1) Opt Express 22(10) 11528 (2019) doi: 10.1364/OE.22.011528; 2) Opt Express 25(22) 27715 (2017) doi: 10.1364/OE.25.027715;  3) Nanomaterials 2022, 12, 4449. https://doi.org/10.3390/nano12244449.
